# Preparation and properties of isocyanate self-healing microcapsule cement-based material

**Yanqing Xian** *

Huangyuan Highway General Section of Qinghai Province, Xining, Qinghai, China

* 2045087410@qq.com

**Data Availability Statement:** All relevant data are within the manuscript and its Supporting Information files.

**Funding:** This research was funded by Qinghai Provincial Transportation Technology Project, grant number HYZD-KT-2023-01, and the APC was

## Abstract

Self-healing microcapsule cement-based materials autonomously repair microcracks within the matrix. This study utilized the interfacial polymerization method to prepare isocyanate microcapsules, incorporating montmorillonite to enhance the wall material. The optimized synthesis conditions were established as follows: a reaction time of 2 hours at 50°C, using 3g of wall material, 13.5g of core material, and 1% montmorillonite, achieving a core material content of 61.5%. Both micro and macro analyses confirm the microcapsules' excellent alkaline resistance and compatibility with cement-based materials. The ideal microcapsule concentration was determined to be 4%, which increased the compressive strength recovery rate of the self-healing mortar to 117.38%.

## Introduction

Cracks inevitably develop in concrete due to external loads, carbonization, temperature fluctuations, and shrinkage reactions, impacting the durability and safety of cement-based materials. The concept of self-healing draws inspiration from the regenerative abilities of natural organisms, where wound repair substances are released post-injury to facilitate healing and restore matrix functionality [1]. Researchers have integrated repair materials within concrete, enabling the release of repair agents through various induction mechanisms upon damage, making self-healing concrete technology a leading method to enhance concrete durability [2].

Recent advancements in self-healing microcapsule cementitious materials have shown significant progress [3]. Microscopic pores within cement-based materials serve as natural storage spaces for microcapsules [4]. Mechanically triggered microcapsules are predominantly studied, where external stimuli cause microcracks in the material, rupturing the microcapsules at crack tips to release core repairing agents, facilitating spontaneous crack repair.

Studies indicate that nanoparticle incorporation can mitigate drawbacks of traditional organic shell microcapsules, such as poor compactness and core material loss [5]. Montmorillonite is a natural mineral silicate and the main mineral component of bentonite. Due to its strong adsorption capacity and good dispersion properties, montmorillonite has a wide range of applications. It possesses many unique properties, especially through inorganic and organic modification, making it widely used as an additive to nano-polymer polymers in the polymer

funded by Qinghai Provincial Department of Transportation. The Qinghai Province Transportation Technology Project provided financial support and site conditions for the project that I presided over and participated in.

**Competing interests:** The authors have declared that no competing interests exist.

materials industry [6,7]. B, W, Jo [8] et al. prepared MMT-UP nanocomposites by inserting UP resin into the silicate layer of MMT composites to improve the performance of polymer concrete. The compressive strength, elastic modulus and splitting tensile strength of polymer concrete using MMT-UP nanocomposite are higher than those using pure UP polymer concrete.Qin [9] et al. found that montmorillonite modified epoxy resin (ER) has better flame retardancy, glass transition and mechanical properties. In this study, montmorillonite was introduced into the porous channel of polyurea shell, with high-activity HDI chosen as the core material to prepare isocyanate self-healing microcapsules with effective sealing properties via interfacial polymerization. Optimization of experimental parameters led to the identification of an ideal preparation process, with subsequent characterization of its fundamental properties. By creating a composite system of microcapsules and cement mortar, the optimal microcapsule dosage was determined [10]. Utilizing the pressure method to induce cracks, the study evaluated the repair impact of microcapsules on mortar's compressive strength and their compatibility with cement-based materials.

## Experiments

### Main raw materials and equipment

The materials and equipment used are detailed in Table 1.

### Microencapsulation synthesis

Polyurea-montmorillonite microcapsules were synthesized using the interfacial polymerization method:

1. Aqueous Phase Preparation: 4.5g of GA was dissolved in 180mL of distilled water at 50˚C with stirring at 500 rpm, followed by the addition of 1.0% montmorillonite and dispersion using an ultrasonic cleaner for 15 minutes.

2. Oil Phase Preparation: 3g of MDI and 13.5g of HDI were mixed thoroughly.

3. Emulsification and Reaction: The oil phase was gradually added to the aqueous phase in a 500mL three-neck flask at room temperature, stirring at 400 rpm to form an emulsion over 45 minutes. The reaction was then continued at 50˚C for 2 hours.

4. Post-reaction Processing: The product was filtered, washed multiple times with distilled water, and dried at 45˚C for 12 hours to obtain the microcapsule powder.

### Preparation of self-healing mortar

The mortar is prepared by first mixing cement, microcapsules, and water at low speed for 30 seconds (Table 2). After adding standard sand, the mixture is blended at high speed for 30 seconds, paused for 90 seconds, and then mixed again at high speed for 60 seconds. The mortar is then poured into a triple mold 40mmx40mmx160mm and compacted using a vibrating table. Excess mortar is removed to level the surface with the mold top. Each mix ratio produces three specimens. These are cured under standard conditions relativehumidity>90 and demolded after 24 hours. Post-demolding, specimens are water-cured at 20±1˚C until the specified age, dried, and then placed in a standard curing room for 7 days after preloading-induced cracking, awaiting self-healing.

Prestress Damage: The specimens were placed under a combined compression and bending tester and loaded with a pre-compression value at a rate of 2.2 to 2.4 kN/s. After reaching the

**Table 1. Raw materials used in the test.**

| The name of the material | | Chemical formula or abbreviation | Specifications |
| --- | --- | --- | --- |
| Microcapsules prepare raw materials | Diphenylmethane diisocyanate | MDI | Analytical pure |
| | Hexamethyl diisocyanate | HDI | Analytical pure |
| | gum arabic | GA | Analytical pure |
| | Triethylenetetramine | TETA | Analytical pure |
| | n-octanol | $C_8H_{18}O$ | Analytical pure |
| | Tween 80 | Tweeen80 | Analytical pure |
| | Polyvinyl alcohol 1788 | PVA | Analytical pure |
| | Polyether F127 | - | Chemically pure |
| | Polyvinylpyrrolidone | PVP | Analytical pure |
| | Sodium dodecylbenzene sulfonate | SDBS | Chemically pure |
| | Sodium lauryl sulfate | SDS | Chemically pure |
| | Nakimontutte | MMT | Chemically pure |
| | Steaming water | $H_2O$ | - |
| Raw materials for testing | cement | - | P·O42.5 |
| | sand | - | Standard sand |
| | water | - | Jirai Water |

preload pressure, the loading was stopped and constant loading condition was maintained for 3 min and then the pressure was unloaded. During this process, due to the inhomogeneity of the material, stress concentrations within the specimen started to form and extend microcracks. The cracks are mainly microcracks with sizes usually ranging from a few micrometres to tens of micrometres, which are mainly distributed on the surface and inside of the specimen, and are mostly surface cracks. In order to describe more accurately the extent of prestress damage, we measured the predetermined pressure values required to induce these cracks. Especially when the specimens were loaded at a loading rate of about 2.3 kN/s and the prestress value reached about 100 MPa, the formation of cracks could be clearly seen. The compressed region was selected for self-healing testing in order to directly compare the difference in performance of the same region before and after self-healing. This helps to quantify the contribution of the self-healing microcapsules to the recovery of material properties and to validate the effectiveness of the self-healing technique.

## Testing and characterization

**Microstructure characterization.** Microcapsule morphology is examined using a SK2009U3 light microscope, ZEISS SUPRA55 scanning electron microscope, and a stereo microscope [11]. Particle sizes are measured with a Malvern Mastersizer2000. The chemical composition is analyzed using a Thermo Scientific Nicolet iS5 Infrared Spectrometer, focusing on characteristic peaks of montmorillonite and microcapsule components, within a wavenumber range of 4000 to 400cm$^{-1}$. Thermal stability of the microcapsules is assessed

**Table 2. Self-healing mortar mix ratiog.**

| Numbering | Cement | Water | Sand | Microcapsules |
| --- | --- | --- | --- | --- |
| MC0 | 450 | 225 | 1350 | 0 |
| MC1 | 450 | 225 | 1350 | 9 |
| MC2 | 450 | 225 | 1350 | 18 |
| MC3 | 450 | 225 | 1350 | 27 |

using a TGA5500 thermogravimeter, with a heating rate of 10˚C/min in an N2 atmosphere up to 800˚C.

**Microcapsule core material content test.** The core material content is determined by grinding the microcapsules, dissolving in acetone, and ultrasonically treating for 1 hour to extract HDI [12,13]. After drying, the capsule wall mass is weighed, and the core material content is calculated using the **Formula 1**:

$$\omega = \frac{W_0 - W_1}{W_0} \times 100\%$$ (1)

where $\omega$ is the core material content, $W_0$ is the initial microcapsule mass, and $W_1$ is the microcapsule wall material mass.

**Characterization of self-healing effects.** The effectiveness of microcapsules in mortar repair is quantified by the compressive strength recovery rate $\eta_m$, calculated as **Formula 2**:

$$\eta_m = \frac{f_m}{f_0} \times 100\%$$ (2)

where $f_0$ is the initial compressive strength, and $f_m$ is the compressive strength post-repair.

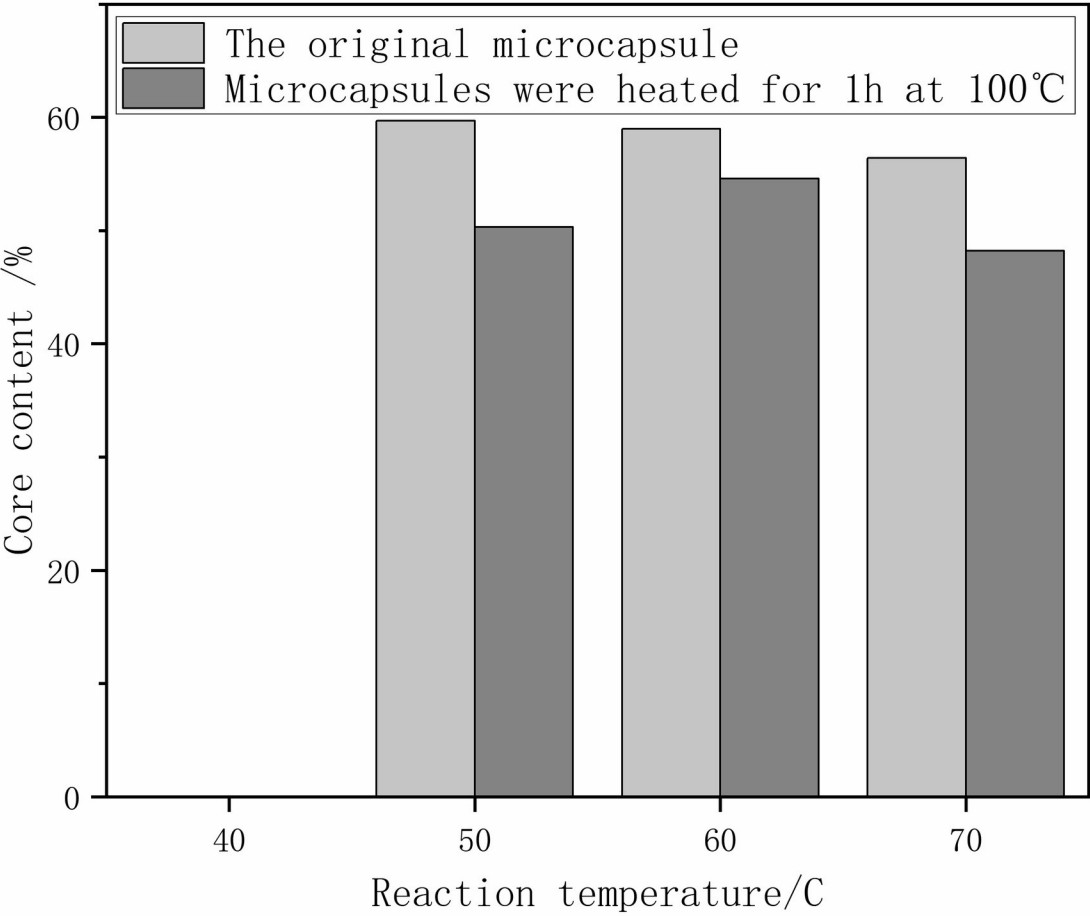

**Fig 1. Core content of microcapsules at different reaction temperatures.**

## Results & discussion

### Optimization of microcapsule synthesis process conditions

**Effect of emulsifier type on spheroidality of microcapsules.** To identify the optimal emulsifier for microcapsule formation, six common emulsifiers were tested under controlled conditions. Despite similar initial results, the addition of TETA significantly enhanced the spheroidality when GA was used as the emulsifier, resulting in uniformly shaped and well-dispersed microcapsules.

**Temperature's impact on microcapsule core content.** Analysis of core content at varying temperatures revealed optimal microcapsule formation at 50°C, with a significant decrease in core content at higher temperatures due to accelerated diffusion and side reactions (**Fig 1**).

As can be seen from Fig 1 (**Fig 1**):

1. When the reaction temperature was 40°C, the product after washing and filtration at the end of the reaction was spherical, but after drying in the oven at 45°C, the microcapsules could not be formed, and the content of the core material at this time was considered as 0. It indicated that the strength of the wall material for forming the microcapsule was very low

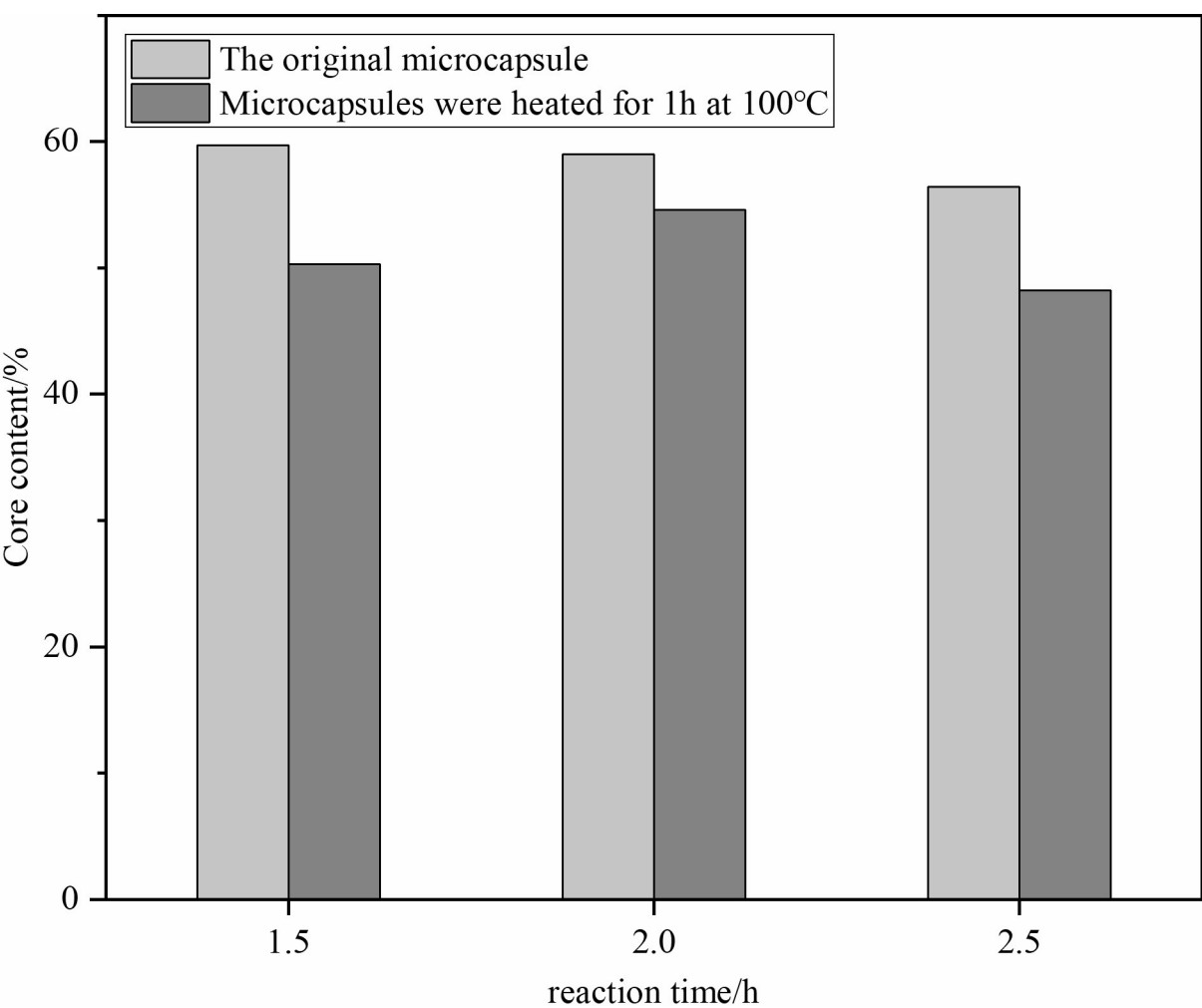

**Fig 2. Core of microcapsules at different reaction times.**

when the reaction temperature was too low, and it was not enough to encapsulate the core material to form the microcapsule.

2. The temperature increased from 50˚C to 70˚C, the core material content of microcapsules showed a decreasing trend, the core material content of microcapsules reached a maximum of 59.0% at 50˚C, and the core material content of microcapsules was the lowest at 70˚C, only 46.0%. The reason for the decrease in the core content of microcapsules is that with the increasing reaction temperature, the side reaction between the diffusion of water into the interior of microcapsules and the core material is accelerated, which reduces the content of the coated core material.

3. The microcapsules dried at different reaction temperatures were dried in an oven at 100˚C for 1 h. The decreasing trend of the microcapsule core content was consistent with the decreasing trend of the dried core content. And with the increase of reaction temperature, the greater the decrease of core material content, indicating that the densification will be reduced beyond a certain temperature.

In summary, the preparation temperature of microcapsules was controlled at about 50˚C.

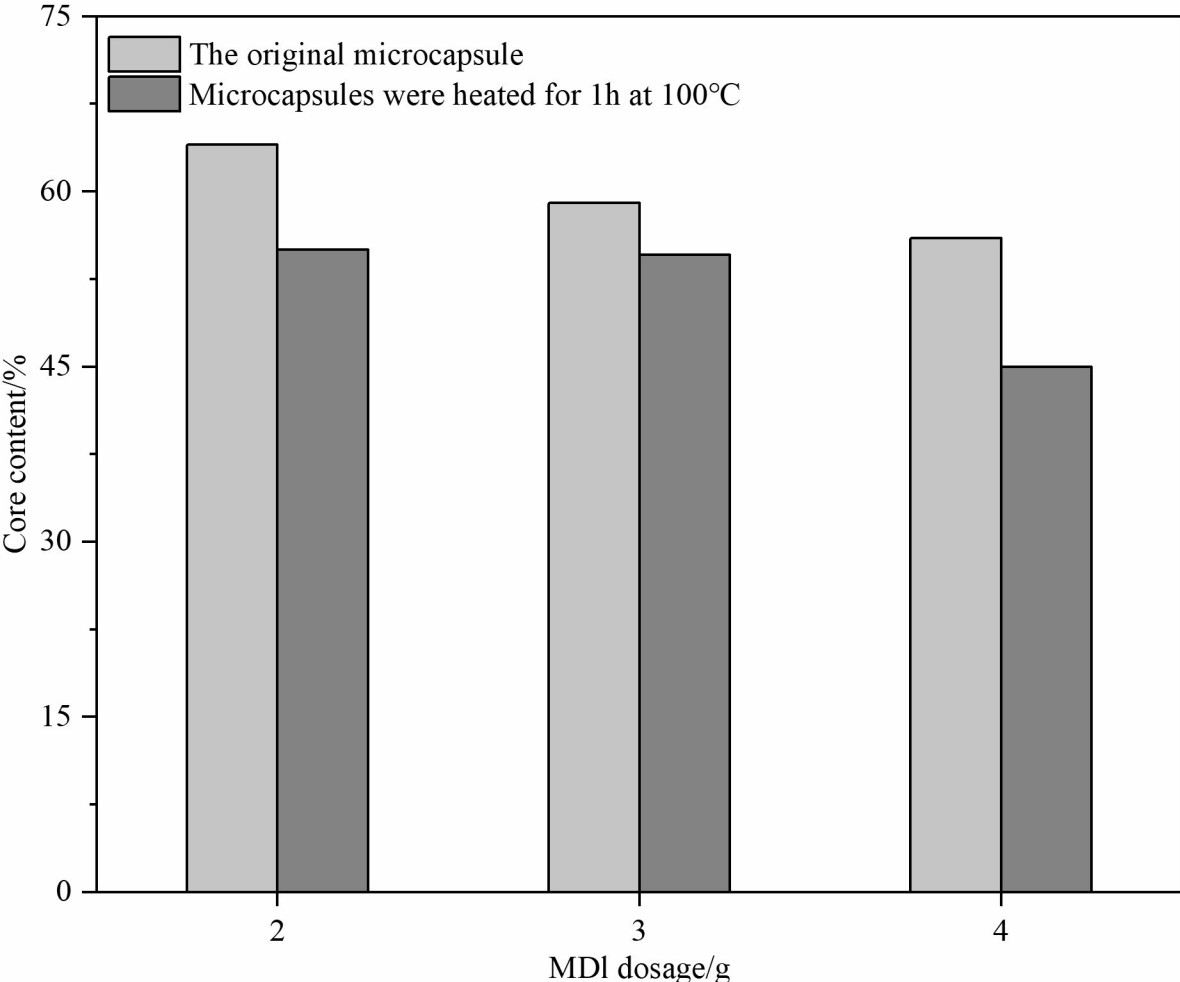

**Fig 3. Effect of MDI dosage on the core content of microcapsules.**

**Influence of reaction time on core content.** Extended reaction times led to a decrease in core content, with the optimal time identified at 2 hours. Notably, microcapsules formed at 1.5 hours showed the highest initial core content but suffered significant losses upon further heating, highlighting the delicate balance between reaction time and microcapsule integrity (**Fig 2**).

**Effect of MDI dosage on microcapsule core content.** Varying the dosage of MDI under constant conditions showed that a moderate dosage of 3g resulted in the highest core retention and optimal shell thickness, balancing durability and efficiency (**Fig 3**).

**Effect of MMT dosage on microcapsule core content.** The influence of MMT dosage on microcapsule core content was analyzed under specific synthesis conditions (**Fig 4**). The core content peaked at 1.0% MMT dosage, achieving 61.5%, indicating optimal microcapsule structure formation at this concentration. Higher MMT levels led to structural degradation and agglomeration, reducing core content. The study established the best synthesis parameters: 2 hours at 50°C, with 3 g of wall material, 13.5 g of core material, and 1.0% MMT.

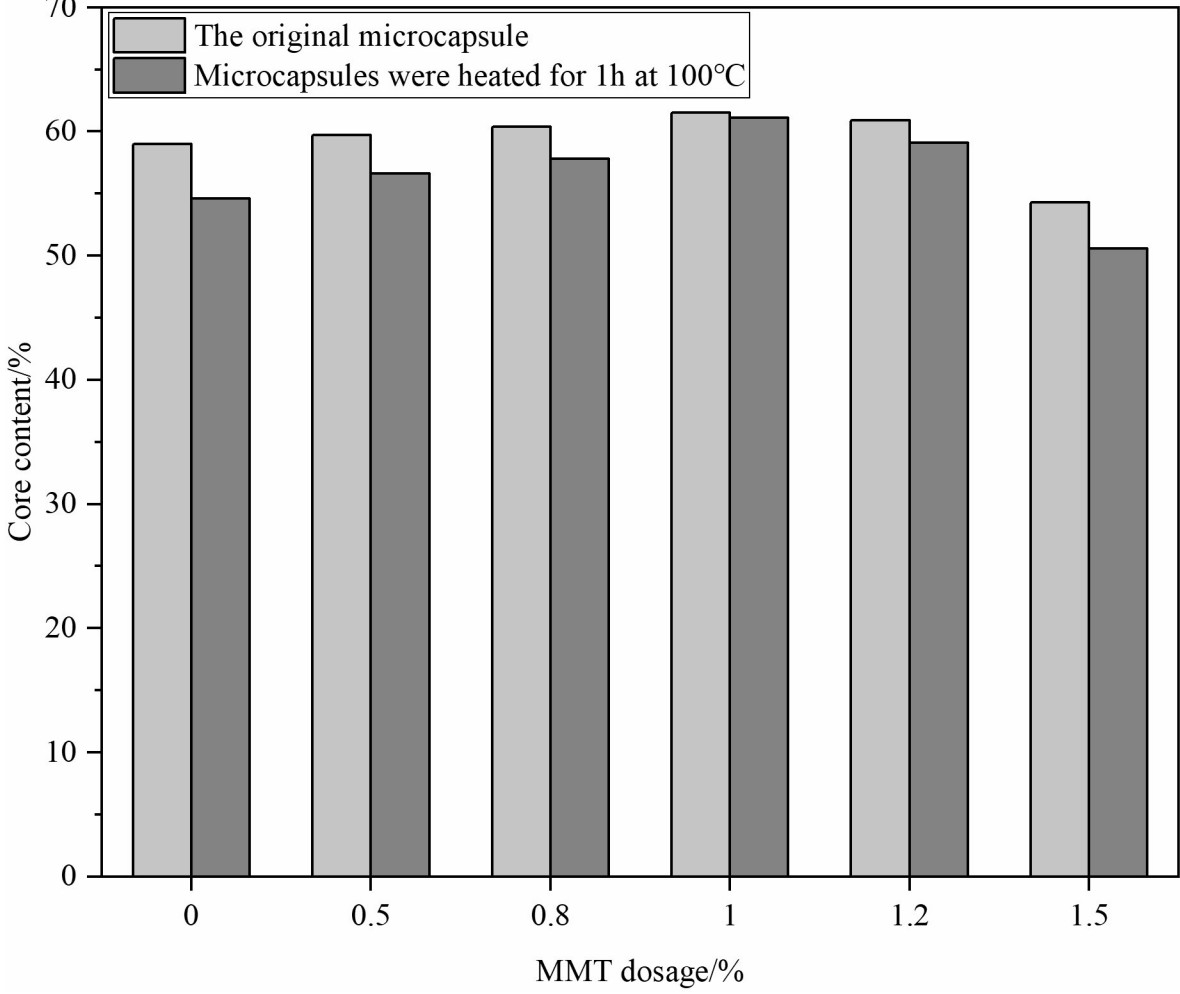

**Fig 4. Effect of MMT dosage on microcapsule core content.**

## Study of microcapsule properties

**Morphology and particle size.**   Microcapsules were mostly spherical, with a uniform particle size distribution primarily between 300–500 $\mu m$ (**Fig 5**), influenced by emulsion speeds in a spherical flask (**Fig 6**). The slightly rough surface of the microcapsules enhances their bonding strength to cementitious materials.

**Infrared spectral analysis.**   Infrared spectra revealed distinct peaks corresponding to the components of microcapsules. Notable peaks included the Al-O and Si-O groups in montmorillonite and the characteristic bands of the polyurea wall material, confirming successful encapsulation of HDI within the polyurea-montmorillonite matrix (**Fig 7**).

In **Fig 7** the peaks at 520 cm-1 correspond to the Al-O expansion vibrations in montmorillonite, while the 1100 cm-1 peaks indicate the Si-O group expansion vibrations. The range of 1500–1300 cm-1 signifies the presence of nanokimontmorillonite. Analysis of the wall material of the microcapsules revealed peaks at 1636 and 1548 cm-1 for the -C = 0 group, 3334 cm-1 for N-H group vibrations, and 2941 and 2855 cm-1 for -CH3 and -CH vibrations, confirming the formation of polyurea walls. Furthermore, the peaks at 520 and 1088 cm-1 suggest the presence of Al-O and Si-O groups in montmorillonite within the wall material, indicating the composition of polyurea and montmorillonite. The infrared spectrum of the microcapsule displayed characteristic peaks of both the core material HDI and the wall material, demonstrating the successful encapsulation of HDI within the polyurea-montmorillonite matrix.

**Thermal stability.**   Starting from **Fig 8**, we observe the core material undergoing weight loss at 106°C, primarily due to moisture evaporation. Subsequently, as the temperature approaches its boiling point, the core material continues to lose weight, culminating in

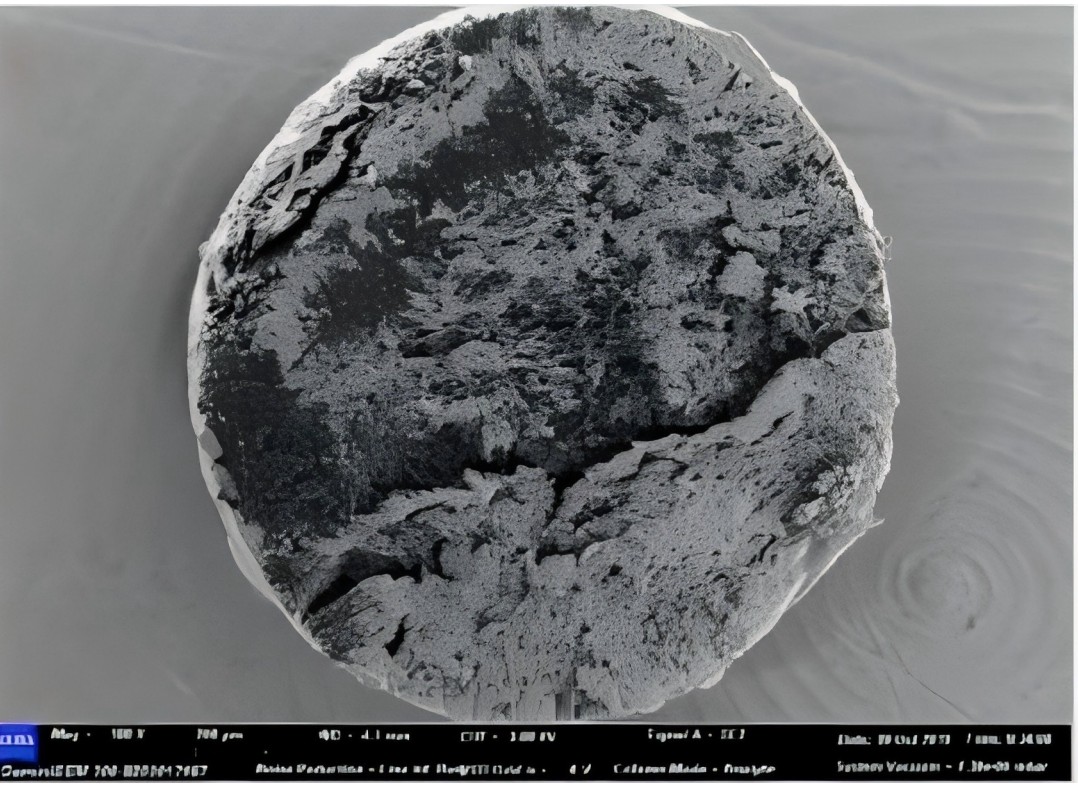

**Fig 5. Microscopic morphology of the microcapsules.**

complete evaporation at approximately 217˚C, resulting in a plateau in the thermogravimetric curve. Concurrently, the wall material initiates weight loss at 217˚C, immediately following the complete weightlessness of the core material. Notably, the mass remains stable around 540˚C. The weight loss of microcapsules initiates at 123˚C, primarily attributed to core material evaporation until reaching 217˚C, transitioning into a plateau phase. Subsequently, from 217 to 540˚C, microcapsules continue to lose weight, mainly due to wall material cracking. Beyond 540˚C, mass loss decelerates or stabilizes, indicating challenges in carbon oxidative decomposition. This indicates the microcapsules' suitability for self-healing concrete applications, as they withstand high temperatures without substantial degradation.

**Alkali resistance.** The efficacy of microcapsules in concrete repair hinges on their ability to withstand and protect the concrete under alkaline conditions. Initially, a saturated calcium hydroxide simulation solution was prepared, followed by adjusting a concrete simulation solution to a pH of 13.4 using a 10% sodium hydroxide solution. The microcapsules were then immersed in the concrete simulation solution for 7 days. Remarkably, the morphology of the microcapsules remained unchanged before and after immersion. They were uniformly and perfectly dispersed at the bottom of the concrete simulation solution, indicating no leakage of the core material and excellent alkali resistance.

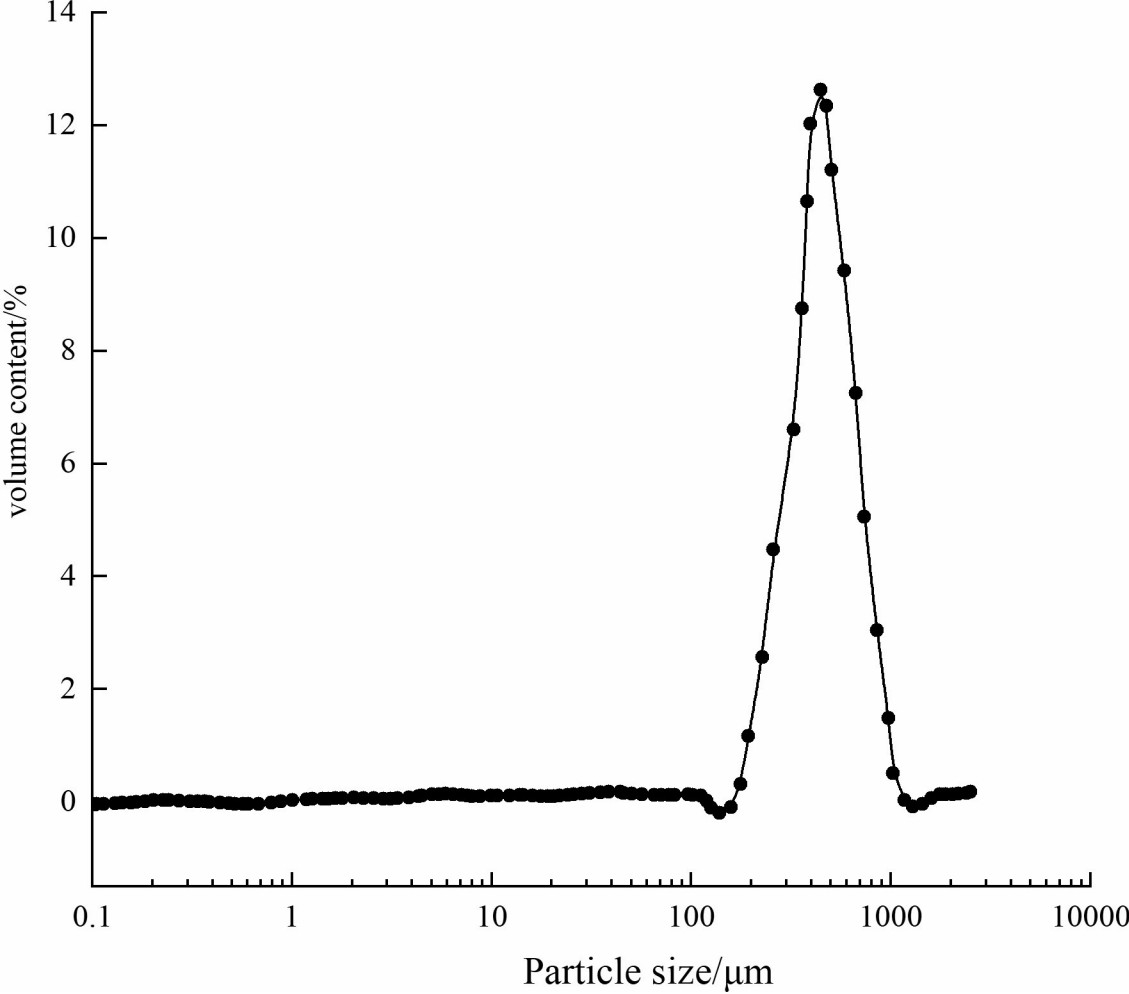

**Fig 6. Particle size distribution of microcapsules.**

### Compatibility analysis of microcapsules and cement slurry experimental

The cement mortar mixed with microcapsules was broken at the end of the fresh moulding, and the microcapsules embedded in the mortar were more intact. Because the degree of hydration of the specimen is very low, the contact surface between the microcapsule and the cement slurry is very weak, and the cracks generated during cracking are not enough to damage the microcapsule. In addition, some hydration products adhered to the microcapsules, which indicated that the microcapsules could survive well in cement mortar.

The microcapsules of 7D age are crushed, and the microcapsules burst into half, and the microcapsules are dispersed evenly, which ensures that there are no obvious fragile parts in the mortar body, and the microcapsules can repair the microcracks produced in the mortar to the highest extent.

### Effect of microcapsule content on the strength of cement mortar

Analysis of Table 3 reveals that the flexural strength of control specimens (without microcapsules) ranges from 121% to 135% of that observed in the 2% to 6% microcapsule groups at the same age. Similarly, the compressive strength of the control specimens is approximately 137% to 168% of that in the microcapsule groups. This discrepancy can be attributed to the relatively

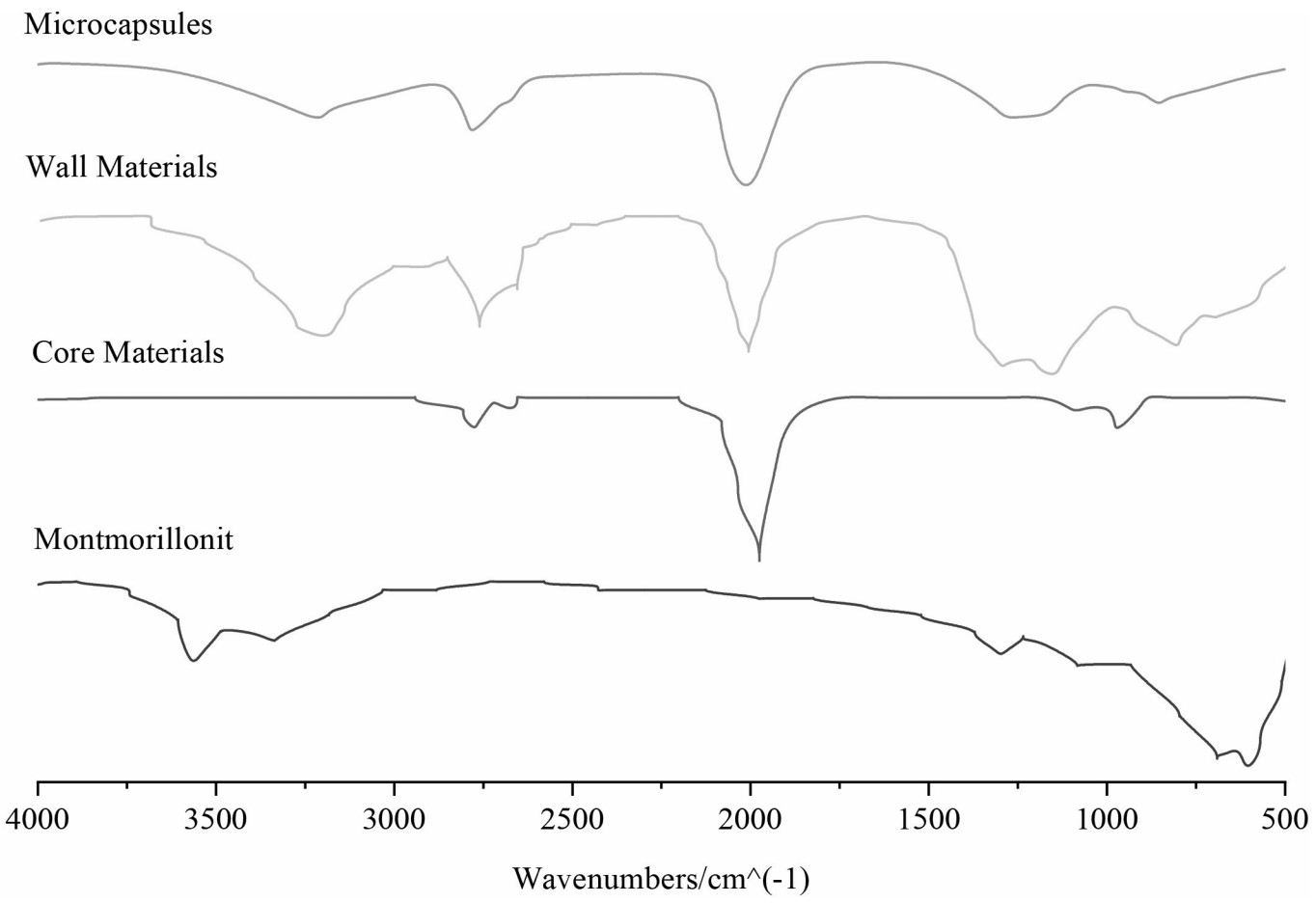

**Fig 7. Infrared spectra of montmorillonite, wall materials, core materials, and microcapsules.**

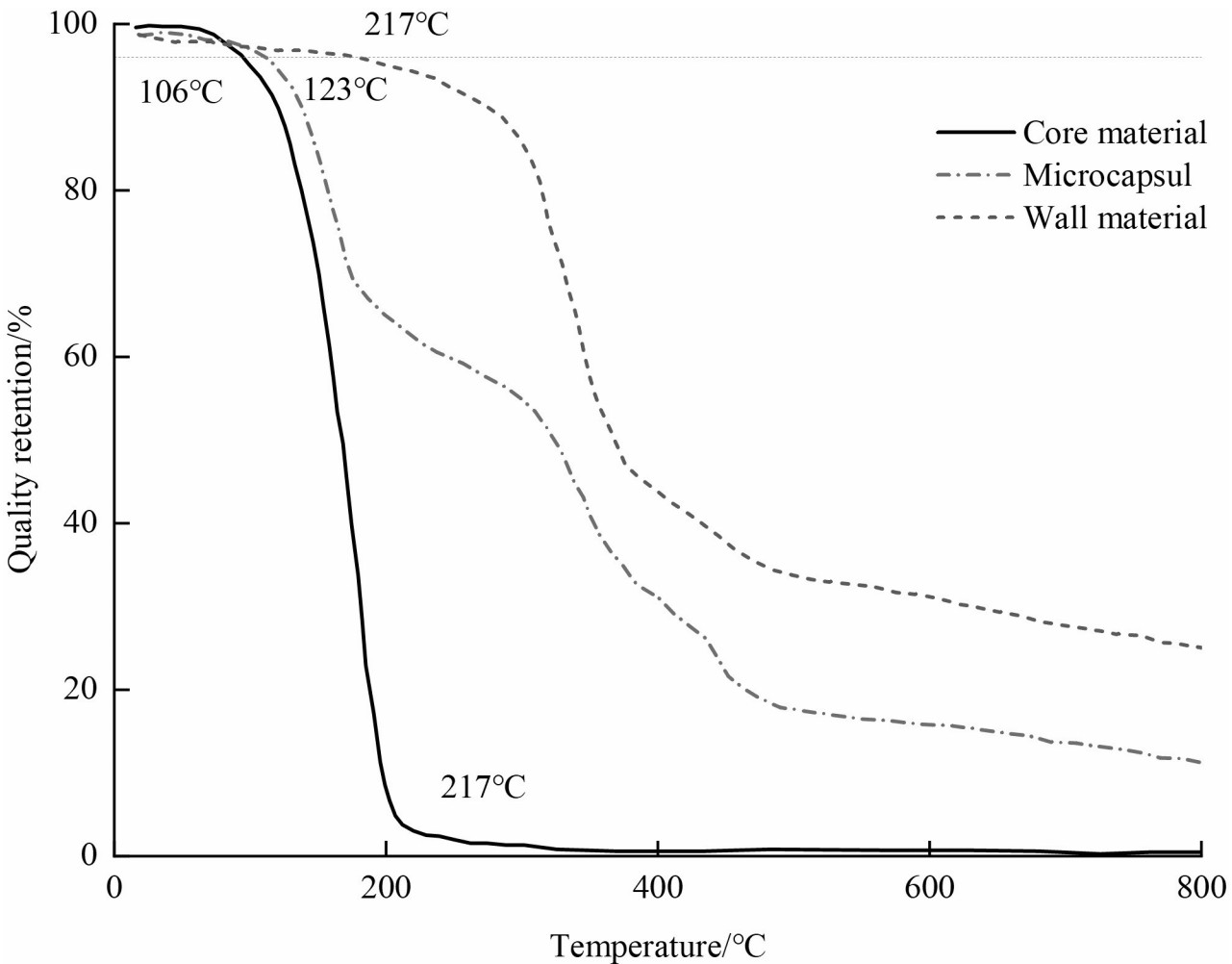

**Fig 8. Thermal weight curve of microcapsules and core materials and wall materials.**

large size of the synthesized microcapsules, which possess lower strength compared to the cement matrix.

## Self-healing effect of cement mortar after microcapsulation

As depicted in Fig 9 the self-healing capability of the mortar diminishes over time, with the strength recovery in the control group showing a nearly linear decline. Conversely, the strength recovery in mortars containing microcapsules tends to stabilize with age. The

**Table 3. Impact of microcapsule content on mortar strength.**

| Microcapsule mixing volume /% | Rupture strength /MPa | | | Compression strength /MPa | | |
|---|---|---|---|---|---|---|
| | 7d | 14d | 28d | 7d | 14d | 28d |
| 0 | 6.55 | 6.97 | 7.46 | 32.43 | 35.97 | 43.23 |
| 2 | 5.24 | 5.62 | 5.96 | 22.56 | 24.76 | 26.81 |
| 4 | 5.01 | 5.85 | 6.18 | 21.64 | 26.35 | 28.66 |
| 6 | 4.86 | 5.43 | 5.64 | 20.04 | 23.69 | 25.70 |

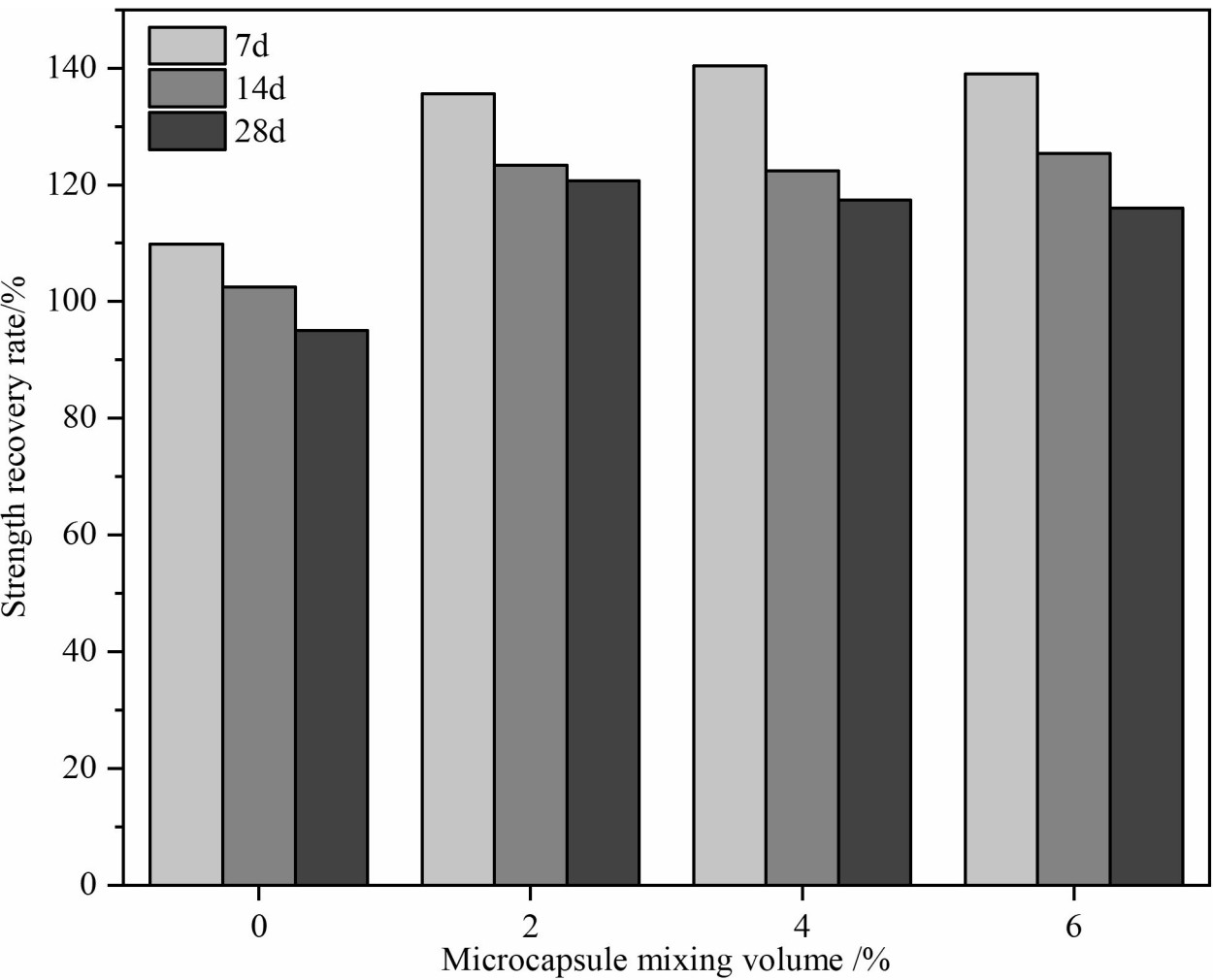

**Fig 9. Strength recovery rate in cement mortar with varying microcapsule dosages post-damage.**

microcapsule content does not significantly influence the recovery rate at the same age. Notably, extending the curing period from 14 to 28 days, particularly in groups with 2% and 4% microcapsule content, markedly slows the decline in strength recovery rate, with an average reduction of only 3.7%.

## Conclusion

1. Microcapsules were synthesized by incorporating montmorillonite (MMT) into the capsule wall through interfacial polymerization. Optimal synthesis conditions were achieved with a reaction time of 2 hours at 50°C, using 3g of wall material and 13.5g of core material, with 1% MMT. This resulted in microcapsules with 61.5% core material content, exhibiting complete particle shapes and good alkali resistance.

2. Microscopic analysis confirmed the robustness of microcapsules during the mortar molding process. The microcapsules remained intact and were only damaged by the expanding cracks within the mortar, demonstrating effective incorporation and durability.

3. The self-healing capability of the mortar diminished over time. However, mortars with microcapsules showed a more stable recovery in strength performance compared to the control group. The optimal microcapsule dosage was determined to be 4%, which achieved a 28-day strength recovery rate of 117.38%.

## Author Contributions

**Writing – review & editing:** Yanqing Xian.

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
