## [Decision Letter · Decision Letter 0]

16 Jun 2024

PONE-D-24-16776Preparation and properties of isocyanate self-healing microcapsule cement-based materialPLOS ONE

Dear Dr. Xian,

Thank you for submitting your manuscript to PLOS ONE. After careful consideration, we feel that it has merit but does not fully meet PLOS ONE’s publication criteria as it currently stands. Therefore, we invite you to submit a revised version of the manuscript that addresses the points raised during the review process.

We look forward to receiving your revised manuscript.

Kind regards,

Jiaolong Ren

Academic Editor

PLOS ONE

Journal Requirements:

"This research was funded by Qinghai Provincial Transportation Technology Project, grant number HYZD-KT-2023-01, and the APC was funded by Qinghai Provincial Department of Transportation."

5. Please ensure that you refer to Figure 2 in your text as, if accepted, production will need this reference to link the reader to the figure.

6. We note you have included a table to which you do not refer in the text of your manuscript. Please ensure that you refer to Table 1 and 2 in your text; if accepted, production will need this reference to link the reader to the Table.

Reviewers' comments:

Reviewer's Responses to Questions

**Comments to the Author**

1. Is the manuscript technically sound, and do the data support the conclusions?

Reviewer #1: Partly

2. Has the statistical analysis been performed appropriately and rigorously? 

Reviewer #1: N/A

3. Have the authors made all data underlying the findings in their manuscript fully available?

Reviewer #1: Yes

4. Is the manuscript presented in an intelligible fashion and written in standard English?

Reviewer #1: Yes

5. Review Comments to the Author

Reviewer #1: 1. The description of the microencapsulation synthesis process in the paper is confusing, as it mentions the interfacial polymerization method but lacks any content related to 'interfacial'.

2. The title for table 2: Table 2. Self-healing mortar mix ratiog. – format error.

3. The explanation of preloading-induced cracking (Prestress Damage) in the text was unclear. The manuscript failed to delineate the mechanisms underlying crack formation and omitted details regarding the characteristics of the cracks, such as size and type (microcracking or surface cracking). Furthermore, it lacked quantitative data on the predefined pressure necessary to induce these cracks.

4. Up to row 89, the manuscript merely references the mass of "wall materials" but does not elucidate the nature or function of these materials within the context of self-healing mortar.

5. The manuscript lacks an explanation regarding the mechanisms and reasons why montmorillonite enhances the wall material. Please reviewe and suppliment the literatures: https://doi.org/10.1016/j.engstruct.2024.117909; https://doi.org/10.1016/j.jclepro.2023.136169; https://doi.org/10.1016/j.conbuildmat.2020.121212

6. Row 73-74: “The compression area is marked for subsequent compressive strength tests post-self-healing.” This statement is ambiguous. There is no description of where the compression area is located. It appears from the manuscript that the entire specimen was subjected to compression. Additionally, the compression was intended to induce cracks, making it unclear why the compression area is designated for post-self-healing tests.

7. Row 101-103: Temperature's impact on microcapsule core content.

(1) It is unclear what "core content" refers to. Does it indicate the core content of the mortar post-self-healing or pre-self-healing?

(2) Only reaction temperatures of 50, 60, and 70 degrees were measured. Why were temperatures of 20, 30, and 40 degrees not tested?

(3) Why was there a significant decrease in core content at higher temperatures, as expected?

8. In Figure 1, the darker column represents microcapsules that were heated for 1 hour at 100°C.

(1) What does 100°C represent? Is it the reaction temperature?

(2) What is the purpose of this comparison column?

9. Row 126: “The slightly rough surface of the microcapsules enhances their bonding strength to cementitious materials.” There was no SEM image showing the morphology of microcapsules.

10. In Figure 7, it is shown that the temperatures at which mass loss initiates for the wall material and the microcapsule are 123°C and 217°C, respectively. However, the manuscript incorrectly describes these two temperatures in reverse.

11. In Figure 7, thermogravimetric (TG) analysis was conducted on the three materials (core, wall, and microcapsule) to obtain their individual mass retention (mass change). However, during this experiment, the three materials were mixed together. It is unclear how the thermal behaviors of the three materials were measured separately under these conditions.

12. Row 161: The cement slurry mixed with microcapsules is crushed at the end of the just forming, and the microcapsules are crushed, the microcapsules embedded in the slurry are intact. The statement "the microcapsules are crushed, the microcapsules embedded in the slurry are intact" is confusing. Based on the content of the paper, all the microcapsules should be embedded in the slurry.

13. In Figure 8, it is shown that mixing 2% microcapsules results in higher strength recovery compared to 0%. However, Table 3 indicates that both the rupture and compressive strength of the mortar mixed with 2% microcapsules are lower than those with 0% microcapsules, contradicting the claim of higher strength recovery.

6. PLOS authors have the option to publish the peer review history of their article (what does this mean?). If published, this will include your full peer review and any attached files.

Reviewer #1: No

---

## [Author Response · Author response to Decision Letter 0]

9 Sep 2024

Hello dear reviewers, Thank you very much for your valuable comments on this manuscript. Please find below my response to your comments for your perusal.

Reviewer's Responses to Questions

Comments to the Author

1. Is the manuscript technically sound, and do the data support the conclusions?

Reviewer #1:Partly

Response：Thank you for reviewing our manuscript and providing valuable comments. We value your feedback and have revised and supplemented the manuscript accordingly to ensure the technical reliability and accuracy of the data supporting the conclusions. Below are our specific responses to the issues you raised:

We have described the experimental methodology in detail to make it easier for readers to understand. We re-analysed the dataset and were able to better support our conclusions. Based on the data provided, we have re-evaluated the presentation of our conclusions to ensure that they are closely related to the data and are not over-interpreted. Our conclusions are now more cautious, reflecting the uncertainties and limitations of the data. You can see the details in our resubmitted manuscript.________________________________________

2. Has the statistical analysis been performed appropriately and rigorously?

Reviewer #1:N/A

Response：Thank you for your valuable comments. In response to your question about the lack of rigour in the execution of the statistical analyses, I have revisited the methods used in the paper and ensured that they are applicable to our study design and data type. Where methods were not applicable, I have made substitutions or adjustments to ensure the accuracy of the analyses. In addition, I have revised the presentation of the results section I have provided thorough answers to your questions in the resubmitted manuscript. You can see the section on this in the resubmitted manuscript.

3. Have the authors made all data underlying the findings in their manuscript fully available?

Reviewer #1:Yes

4. Is the manuscript presented in an intelligible fashion and written in standard English?

Reviewer #1:Yes

5. Review Comments to the Author

Reviewer #1:

1. The description of the microencapsulation synthesis process in the paper is confusing, as it mentions the interfacial polymerization method but lacks any content related to 'interfacial'.

Response：Interfacial polymerisation is a method in which an emulsifier is used to form a water/oil or oil/water emulsion, and the monomers from the polymerisation reaction form a polymer film on the surface of the core material and gradually form a capsule shell, and finally the capsule is separated from the water or oil phase. The term "interface" refers to the irreversible polycondensation reaction at the interface of two liquid phases. It is therefore not described in the manuscript, but the preparation process is described in the body of the manuscript.

2. The title for table 2: Table 2. Self-healing mortar mix ratiog. – format error.

Response：We have made changes based on the comments you have made.You can see it on line 80 of the resubmitted manuscript.

3. The explanation of preloading-induced cracking (Prestress Damage) in the text was unclear. The manuscript failed to delineate the mechanisms underlying crack formation and omitted details regarding the characteristics of the cracks, such as size and type (microcracking or surface cracking). Furthermore, it lacked quantitative data on the predefined pressure necessary to induce these cracks.

Response：In response to your questions, I have revised the relevant paragraphs and added the necessary information. The following are the modifications:“The specimens were placed under a combined compression and bending tester and loaded with a pre-compression value at a rate of 2.2 to 2.4 kN/s. After reaching the preload pressure, the loading was stopped and constant loading condition was maintained for 3 min and then the pressure was unloaded. During this process, due to the inhomogeneity of the material, stress concentrations within the specimen started to form and extend microcracks. The cracks are mainly microcracks with sizes usually ranging from a few micrometres to tens of micrometres, which are mainly distributed on the surface and inside of the specimen, and are mostly surface cracks. In order to describe more accurately the extent of prestress damage, we measured the predetermined pressure values required to induce these cracks. Especially when the specimens were loaded at a loading rate of about 2.3 kN/s and the pre-stress value reached about 100 MPa, the formation of cracks could be clearly seen.”

You can see it on line 81 of the resubmitted manuscript.

4. Up to row 89, the manuscript merely references the mass of "wall materials" but does not elucidate the nature or function of these materials within the context of self-healing mortar.

Response：In response to your question, we have made the following changes and explanations:Prior to line 89, we have added a description detailing the role and importance of the cited montmorillonites in self-healing mortars. To further support our argument, we cite several research papers on the application of similar wall materials in self-healing materials to demonstrate the effectiveness and potential of these materials in practical applications.You can see this on line 43 of the manuscript.

5. The manuscript lacks an explanation regarding the mechanisms and reasons why montmorillonite enhances the wall material. Please reviewe and suppliment the literatures: https://doi.org/10.1016/j.engstruct.2024.117909; https://doi.org/10.1016/j.jclepro.2023.136169; https://doi.org/10.1016/j.conbuildmat.2020.121212.

Response：An explanation of the mechanisms and causes of montmorillonite-enhanced walls has been added, as detailed in line 43 of the resubmitted manuscript, with additional literature [6-9].

6. Row 73-74: “The compression area is marked for subsequent compressive strength tests post-self-healing.” This statement is ambiguous. There is no description of where the compression area is located. It appears from the manuscript that the entire specimen was subjected to compression. Additionally, the compression was intended to induce cracks, making it unclear why the compression area is designated for post-self-healing tests.

Response：With regard to "marking of compression zones for subsequent compressive strength testing after self-healing" as stated in lines 73-74, there is indeed a lack of clarity in the statement. In order to solve this problem, we will clarify the following points in the revised draft:

1. We use the pre-compression method to pre-crack the specimen, and the pre-compression process is described in the manuscript, as you understand it to be compression of the entire specimen.

2.It is true that compression is used to induce cracks in these regions. The compressed regions were chosen for post self-healing compressive strength testing in order to directly compare the difference in performance of the same regions before and after self-healing. This helps to quantify the contribution of the self-healing microcapsules to the recovery of material properties and to validate the effectiveness of the self-healing technique.

To improve the clarity of the presentation, we have added an explanatory paragraph here to explain why the compression region was designated for the self-healing post-test.

Thank you for your valuable comments, and we believe that these revisions will make the paper better and easier to understand. Below is the revised content "Prestress Damage: The specimens were placed under a combined compression and bending tester and loaded with a pre-compression value at a rate of 2.2 to 2.4 kN/s. After reaching the preload pressure, the loading was stopped and constant loading condition was maintained for 3 min and then the pressure was unloaded. During this process, due to the inhomogeneity of the material, stress concentrations within the specimen started to form and extend microcracks. The cracks are mainly microcracks with sizes usually ranging from a few micrometres to tens of micrometres, which are mainly distributed on the surface and inside of the specimen, and are mostly surface cracks. In order to describe more accurately the extent of prestress damage, we measured the predetermined pressure values required to induce these cracks. Especially when the specimens were loaded at a loading rate of about 2.3 kN/s and the pre-stress value reached about 100 MPa, the formation of cracks could be clearly seen.The compressed region was selected for self-healing testing in order to directly compare the difference in performance of the same region before and after self-healing. This helps to quantify the contribution of the self-healing microcapsules to the recovery of material properties and to validate the effectiveness of the self-healing technique."

You can also read this section on line 81 of the resubmitted manuscript.

7. Row 101-103: Temperature's impact on microcapsule core content

(1) It is unclear what "core content" refers to. Does it indicate the core content of the mortar post-self-healing or pre-self-healing?

Response：Both are included here because we measure and analyse the change in microencapsulated core content after heating.

(2) Only reaction temperatures of 50, 60, and 70 degrees were measured. Why were temperatures of 20, 30, and 40 degrees not tested?

Response：During the test, we found that when the reaction temperature was 40 ℃, the product after washing and filtration at the end of the reaction was spherical, but after drying in the oven at 45 ℃, it could not be formed into a microcapsule, and the content of the core material at this time was regarded as 0. This indicates that the strength of the wall material for forming the microcapsules at the reaction temperature was too low, and it was not enough to encapsulate the core material to form a microcapsule.

(3) Why was there a significant decrease in core content at higher temperatures, as expected?

Response：The microcapsule core content decreases because as the reaction temperature continues to rise, it accelerates the side reaction that occurs between the diffusion of water into the interior of the microcapsule and the core material, which reduces the content of the core material that is encapsulated.

Response：We have re-added this section to lines 122-138 of the manuscript.

8. In Figure 1, the darker column represents microcapsules that were heated for 1 hour at 100°C.

(1) What does 100°C represent? Is it the reaction temperature?

(2) What is the purpose of this comparison column?

Response：During this step of the test, we were drying the microcapsules after drying at different reaction temperatures in an oven at 100°C for 1 h. The purpose was to observe the decreasing trend of the core content of the microcapsules and the decreasing trend of the core content after drying.

Response：We have re-added this section to lines 122-138 of the manuscript.

9. Row 126: “The slightly rough surface of the microcapsules enhances their bonding strength to cementitious materials.” There was no SEM image showing the morphology of microcapsules.

Response：We have added Fig. 5 Microscopic morphology of microcapsules based on your comments.

10. In Figure 7, it is shown that the temperatures at which mass loss initiates for the wall material and the microcapsule are 123°C and 217°C, respectively. However, the manuscript incorrectly describes these two temperatures in reverse.

Response：Thank you for your comments, here it is due to a deviation in the translation of the drawing, there is no problem with the content of the manuscript. Figure 7 has been amended.

11. In Figure 7, thermogravimetric (TG) analysis was conducted on the three materials (core, wall, and microcapsule) to obtain their individual mass retention (mass change). However, during this experiment, the three materials were mixed together. It is unclear how the thermal behaviors of the three materials were measured separately under these conditions.

Response：Here we have mainly used thermal stability analysis to analyse the thermal behaviour of materials in microcapsules. At the beginning of the experiment, we had a general idea of the temperature at which the weight loss behaviour of each material occurs.

12. Row 161: The cement slurry mixed with microcapsules is crushed at the end of the just forming, and the microcapsules are crushed, the microcapsules embedded in the slurry are intact. The statement "the microcapsules are crushed, the microcapsules embedded in the slurry are intact" is confusing. Based on the content of the paper, all the microcapsules should be embedded in the slurry.

Response：This was mistranslated, and we have amended it to read ‘The cement mortar with microcapsules is broken at the end of the fresh moulding, and the microcapsules embedded in the mortar are more intact.’ You can see this in line 195 of the resubmitted manuscript.

13. In Figure 8, it is shown that mixing 2% microcapsules results in higher strength recovery compared to 0%. However, Table 3 indicates that both the rupture and compressive strength of the mortar mixed with 2% microcapsules are lower than those with 0% microcapsules, contradicting the claim of higher strength recovery.

Response：Fig. 8 shows the strength recovery after damage repair of cement mortar with different microcapsule dosage. Table 3 Effect of microcapsule dosing on strength of cement mortar. The meanings of the two expressions are different. And we explained in the manuscript that this is due to the larger size of the microcapsules synthesised in this study and the lower strength of the microcapsules themselves compared to the cement matrix. Finally, we got the conclusion that ‘in order to minimise the strength damage of the microcapsules on the mortar, but also to make the mortar have a certain self-healing effect, the dosage of microcapsules was controlled at 4%, in which the strength recovery rate at 28 d could reach 117.38%’.

---

## [Decision Letter · Decision Letter 1]

20 Sep 2024

PONE-D-24-16776R1Preparation and properties of isocyanate self-healing microcapsule cement-based materialPLOS ONE

Dear Dr. Xian,

Thank you for submitting your manuscript to PLOS ONE. After careful consideration, we feel that it has merit but does not fully meet PLOS ONE’s publication criteria as it currently stands. Therefore, we invite you to submit a revised version of the manuscript that addresses the points raised during the review process.

We look forward to receiving your revised manuscript.

Kind regards,

Jiaolong Ren

Academic Editor

PLOS ONE

Reviewers' comments:

Reviewer's Responses to Questions

**Comments to the Author**

1. If the authors have adequately addressed your comments raised in a previous round of review and you feel that this manuscript is now acceptable for publication, you may indicate that here to bypass the “Comments to the Author” section, enter your conflict of interest statement in the “Confidential to Editor” section, and submit your "Accept" recommendation.

Reviewer #1: (No Response)

2. Is the manuscript technically sound, and do the data support the conclusions?

Reviewer #1: (No Response)

3. Has the statistical analysis been performed appropriately and rigorously? 

Reviewer #1: (No Response)

4. Have the authors made all data underlying the findings in their manuscript fully available?

Reviewer #1: (No Response)

5. Is the manuscript presented in an intelligible fashion and written in standard English?

Reviewer #1: (No Response)

6. Review Comments to the Author

Reviewer #1: (No Response)

7. PLOS authors have the option to publish the peer review history of their article (what does this mean?). If published, this will include your full peer review and any attached files.

Reviewer #1: No

---

## [Author Response · Author response to Decision Letter 1]

4 Nov 2024

I followed the editor's instructions and the editor asked me to provide a revised version directly

---

## [Editor Report · Decision Letter 2]

15 Nov 2024

Preparation and properties of isocyanate self-healing microcapsule cement-based material

PONE-D-24-16776R2

Dear Dr. Xian,

We’re pleased to inform you that your manuscript has been judged scientifically suitable for publication and will be formally accepted for publication once it meets all outstanding technical requirements.

Kind regards,

Jiaolong Ren

Academic Editor

PLOS ONE
---

## [Editor Report · Acceptance letter]

19 Nov 2024

PONE-D-24-16776R2 

PLOS ONE

Dear Dr. Xian, 

I'm pleased to inform you that your manuscript has been deemed suitable for publication in PLOS ONE. Congratulations! Your manuscript is now being handed over to our production team.

Kind regards, 

on behalf of

Dr. Jiaolong Ren 

Academic Editor

PLOS ONE